# Enantioselective Recognition of L-Lysine by ICT Effect with a Novel Binaphthyl-Based Complex

**DOI:** 10.3390/mi14030500

**Published:** 2023-02-21

**Authors:** Shi Tang, Zhaoqin Wei, Jiani Guo, Xiaoxia Sun, Yu Hu

**Affiliations:** 1Jiangxi Key Laboratory of Organic Chemistry, Jiangxi Science and Technology Normal University, Nanchang 330013, China; 2College of Chemistry, Nanchang University, Nanchang 330031, China

**Keywords:** enantioselective recognition, ICT effect, intermolecular hydrogen bond

## Abstract

A novel triazole fluorescent sensor was efficiently synthesized using binaphthol as the starting substrate with 85% total end product yield. This chiral fluorescence sensor was proved to have high specific enantioselectivity for lysine. The fluorescence intensity of *R*-**1** was found to increase linearly when the equivalent amount of L-lysine (0–100 eq.) was gradually increased in the system. The fluorescence intensity of L-lysine to *R*-**1** was significantly enhanced, accompanied by the red-shift of emission wavelength (389 nm to 411 nm), which was attributed to the enhanced electron transfer within the molecular structure, resulting in an *ICT* effect, while the fluorescence response of D-lysine showed a decreasing trend. The enantioselective fluorescence enhancement ratio for the maximum fluorescence intensity was 31.27 [ef = |(*I_L_* − *I*_0_)/(*I_D_* − *I*_0_)|, 20 eq. Lys], thus it can be seen that this fluorescent probe can be used to identify and distinguish between different configurations of lysine.

## 1. Introduction

In recent years, fluorescent sensors have gained more attention from researchers due to their better sensitivity and short recognition response time, good selectivity, and ease of operation. Among many recognition applications, the enantiomeric fluorescence recognition of chiral organic compounds has gradually become a research hotspot for fluorescent probe design [1,2,3,4,5,6]. A literature review reveals that highly enantioselective fluorescent sensors for recognizing chiral molecules (e.g., amino alcohols and amino acids and related chiral small molecule analogs) are also being developed [7,8]. Chiral molecules are also receiving increasing attention in materials composition, organic synthesis, and agricultural products.

Amino acids are used as nucleophilic reagents in various reactions occurring at carbon–carbon bonds [9]. Amino acids are also an indispensable part of the human body. In the human body, amino acids are important raw materials for protein synthesis and are equally important for human development, normal metabolism, and the continuation of life [10]. Lysine is a fatty amino acid and one of the basic essential amino acids for humans and mammals [11,12,13,14]. The body cannot synthesize it by itself, and it must be supplemented with food. L-Lys can promote skeletal muscle growth [15,16], enhance the body’s immunity, resist the virus, promote fat oxidation, relieve anxiety, etc. It also promotes the absorption of certain nutrients and can work synergistically with some nutrients to improve the performance of various nutrients. Synergistic effects can better express the physiological functions of various nutrients [17,18]. The deficiency of D-Lys can lead to adverse effects such as uremia. The absorption efficiency of D-Lys and L-Lys is different, and D-Lys can hardly be absorbed and utilized, while the biological activity is mainly L-Lys. Lysine is essential for the function of proteins. As a drug target, lysine is not only versatile but also reliable [19]. Therefore, it is also a crucial step for differentiating the enantiomers of amino acids.

Among them, chiral BINOL belongs to the C_2_ axis chiral due to its unique steric rigidity, multiple modification sites, stable chemical properties, and good fluorescence properties. Therefore, chiral binaphthols can be used as substrates for various types of fluorescent probes, which can present good enantioselectivity for various amino acids and their derivatives, amino alcohols, and razole-type drugs [20]. It is because of these advantages that we chose it as a starting substrate, and in order to increase the fluorescence recognition group, we introduced the triazole group, so the compound was designed, synthesized, and studied.

In this article, novel chiral fluorescent probes based on binaphthol are introduced, and the corresponding experimental study is carried out. It was found that the probe had a specific recognition of lysine for common enantiomeric amino acid recognition, where it showed a fluorescence enhancement and red-shifted emission wavelength trend for L-Lys, while D-Lys showed a decreasing trend. The enantiomeric fluorescence enhancement ratio was 31.27.

## 2. Materials and Methods

Shanghai Aladdin provided all the reagents used in this experiment, and the chemicals used were purchased from the corresponding suppliers or synthesized by known routes. ^1^H NMR and ^13^C NMR were measured on the Bruker AM-400WB spectrometer, the instrument used for fluorescence experiments was a Hitachi F-7100 Fluorescence spectrophotometer, and melting points were determined using an X-4 melting point tester. The optical rotation was performed by a Rudolph AUTOPOL IV automatic polarimeter in chromatographic methanol, the ESI-MS spectral data were measured by a Bruker Amazon ion-trap mass spectrometer, and the nonlinear curve fitting software was by Origin 2021.

### 2.1. Experimental

#### 2.1.1. *R*-3,3’-bis((Prop-2-yn-1-yloxy) methyl)-[1,1’-binaphthalene]-2,2’-diol. (*R*-c)

*R*-3,3’-bis(hydroxymethylene)-2,2’-bis(methoxy-methoxy)-1,1’-binaphthyl (*R*-b) (2.0 g, 4.6 mmol) with 10 mL of ultra-dry THF was added to a 50 mL single-mouth flask, then add sodium hydride (0.49 g, 20.2 mmol) in batches and continue the reaction for 2 h, then slowly add 3-bromopropyne (1.8 mL, 23.0 mmol) to the reaction system. Allow the reaction to be overnight after dropping. After the detection reaction of thin layer chromatography spot plate (TLC), the reaction was pretreated to obtain 1.8 g of yellow oily liquid, Y = 76.6%. Under the condition of a turning ice bath, the product (1.8 g, 1.5 mmol) was added to a 100 mL single-mouth bottle and dissolved with 20 mL of CH_3_OH/THF (CH_3_OH:THF = 2:3) solution, and after 10 minutes, we continued to slowly add 5 mL of concentrated hydrochloric acid dropwise, and allowed it to continue to react for 3 hours. After the thin-layer chromatography spot plate (TLC) detection reaction was completed, the yellow oily liquid was obtained by spin drying by silica gel column chromatography (unfolding agent petroleum ether:ethyl acetate = 3:1), Y = 72.5 %. ^1^H NMR (400 MHz, CDCl_3_) *δ* 7.96 (s, 2H), 7.87 (d, J = 8.0 Hz, 2H), 7.35 (t, J = 7.4 Hz, 2H), 7.32–7.21 (m, 2H), 7.11 (d, J = 8.3 Hz, 2H), 6.01 (s, 2H), 4.96 (s, 4H), 4.35 (d, J = 2.2 Hz, 4H), 2.51 (s, 2H). ^13^C NMR (101MHz, CDCI_3_) *δ* 150.05, 132.49, 128.46, 127.97, 127.29, 126.12, 124.45,123.4, 123.04, 112.11, 78.25, 74.21, 67.69, 56.87 (ppm).

#### 2.1.2. Synthesis of Probe *R*-**1**

*R*-3,3’-bis((prop-2-yn-1-yloxy) methyl)-[1,1’-binaphthalene]-2,2’-diol (*R*-c) (1.0 g, 2.26 mmol) and 2-azidoacetate (0.5 mL, 5.4 mmol) were added to a 100 mL aubergine flask with 20 mL of tetrahydrofuran. After the reaction system was stirred for 5 min at 0 °C, sodium ascorbate (0.98 g, 4.9 mmol), copper sulfate pentahydrate (0.56 g, 2.2 mmol), and 8 mL of water were injected into the reaction system overnight. After TLC detected the complete reaction, the reaction was quenched in ice water; dichloromethane was extracted three times, washed once with saturated salt water, and dried with anhydrous magnesium sulfate. Filter and silica gel column chromatography (unfolding agent petroleum ether:ethyl acetate = 1:2) obtained a white solid 1.25 g, Y = 85.0%. M.P. 65–67 °C. [$\alpha {]}_{D}^{25}$ 75 (c = 1, CH_3_OH). MS-ESI m/z: calcd for [C_34_H_32_N_6_O_8_+H]^+^ 653.2360; found 653.2395. ^1^H NMR (400 MHz, CDCl_3_) *δ* 7.90 (s, 2H), 7.85 (d, *J* = 8.1 Hz, 2H), 7.69 (s, 2H), 7.33 (t, *J* = 7.4 Hz, 2H), 7.25 (dd, *J* = 15.2, 7.1 Hz, 2H), 7.10 (d, *J* = 8.4 Hz, 2H), 6.70–6.47 (m, 2H), 5.08 (s, 4H), 4.95 (ABq, ∆δ_AB_ = 0.07, *J* = 12.5 Hz, 4H), 4.84 (s, 4H), 3.74 (s, 6H). ^13^C NMR (101 MHz, CDCl_3_) *δ* 166.65, 151.35, 145.10, 133.64, 129.33, 128.79, 128.16, 126.90, 125.76, 124.54, 124.20, 123.81, 113.83, 69.70, 63.90, 52.95, 50.62 (ppm).

#### 2.1.3. Synthesis of Probe *S*-**1**

*S*-3,3’-bis((prop-2-yn-1-yloxy) methyl)-[1,1’-binaphthalene]-2,2’-diol (*S*-c) (0.30 g, 0.67 mmol) and 2-azidoacetate (0.16 mL, 1.60 mmol) were added to a 100 mL aubergine flask with 10 mL of tetrahydrofuran. After the reaction system was stirred for 5 min at 0 °C, sodium ascorbate (0.30 g, 1.50 mmol), copper sulfate pentahydrate (0.17 g, 0.67 mmol), and 6 mL of water were injected into the reaction system overnight. After TLC detected the complete reaction, the reaction was quenched in ice water; dichloromethane was extracted three times, washed once with saturated salt water, and dried with anhydrous magnesium sulfate. Filter and silica gel column chromatography (unfolding agent petroleum ether:ethyl acetate = 1:2) obtained a white solid 0.38 g, Y = 86.7 %. M.P. 62–63 °C. [$\alpha {]}_{D}^{25}$-75 (c = 1, CH_3_OH). MS-ESI m/z: calcd for [C_34_H_32_N_6_O_8_+H]^+^ 653.2360; found 653.2395. ^1^H NMR (400 MHz, CDCl_3_) *δ* 7.86 (s, 2H), 7.82 (d, J = 7.9 Hz, 2H), 7.66 (d, J = 1.8 Hz, 2H), 7.31 (t, J = 6.9 Hz, 2H), 7.22 (t, J = 7.6 Hz, 2H), 7.08 (d, J = 8.2 Hz, 2H), 6.67 (s, 2H), 5.05 (d, J = 1.9 Hz, 4H), 4.91 (ABq, ∆δ_AB_ = 0.06, J = 12.5 Hz, 4H), 4.81 (s, 4H), 3.72 (d, J = 2.1 Hz, 6H). ^13^C NMR (101 MHz, CDCl_3_) *δ* 166.68, 151.34, 145.05, 133.61, 129.27, 128.75, 128.17, 126.91, 125.68, 124.54, 124.23, 123.83, 113.83, 69.74, 63.86, 53.01, 50.61. (ppm).

#### 2.1.4. Preparation of Solutions Required for the Test

A 65 mg probe *R*-**1** and a 65 mg probe *S*-**1**, respectively, were placed into two 10 mL volume volumetric flasks; add chromatographic methanol to dissolve and quantify to 10 mL. At this time, the concentration of the test mother liquor is 0.1 M, and then the mother liquor is diluted to a concentration of 2.0 × 10^−5^ mol L^−1^, which needs to be ready to be used at any time.

The commonly used amino acids (D/L-cysteine, D/L-phenylalanine, D/L-alanine, D/L-methionine, D/L-proline, D/L-lysine, D/L-glutamic acid, D/L-glutamine, D/L-arginine, D/L-serine, D/L-threonine, D/L- asparagine, D/L-aspartic acid, D/L-valine, D/L-histidine, D/L-tryptophan, D/L-leucine D/L-tyrosine) were used to configure the above amino acid solution at a concentration of 0.1 M in deionized water, which needs to be prepared and used freshly. At room temperature of 25 °C, 2 mL of *R*-**1** test solution was added to 3.5 mL of high-transparency quartz fluorescence cuvette, followed by 8 μL of amino acids to be measured, and then the fluorescence spectral response was carried out, λ_ex_ = 318 nm, slits = 5.0/2.5 nm, and the fluorescence response time was 0.8 s.

## 3. Results and Discussion

### 3.1. Synthesis Step Scheme

Figure 1 shows the synthesis procedure for probes *R*-**1** and *S*-**1**. BINOL fluorescent probes *R*-**1** and *S*-**1** modified by the triazole group were synthesized using dinaphthol as the starting substrate. Propyne and derivatives *R*-c and *S*-c were synthesized by addition and substitution reactions, and then the probes were synthesized by Click reaction and methyl azidoacetate with THF as solvent at room temperature. The yield is as high as more than 80%. The synthesized compounds were validated by ^1^H NMR, ^13^C NMR, and ESI-MS.

### 3.2. Fluorescence Studies

#### 3.2.1. Fluorescence Studies of Lysine

Fluorescence response of the *R*-**1** probe to thirty-six amino acids (eighteen pairs of amino acid isomers) is as noted below. The solution of 0.1 M of amino acids was configured in deionized water, and *R*-**1** was configured in chromatographic methanol at a concentration of 2.0 × 10^−5^ M. During the test, each pair of amino acid isomers (20 eq.) was added separately to the test master solution *R*-**1** for fluorescence response testing.

As shown in Figure 1a, only L-Lys showed a significant fluorescence intensity change after adding thirty-six amino acids to the test system. As show in the figure, L-Lys significantly improved the fluorescence response of the *R*-**1** probe with an *I/I*_0_ value of 1.79. Then, the following research studies were conducted by examining the fluorescence response of the lysine enantiomers. In the chromatographic CH_3_OH system, when L-Lys (20 fold equivalent) was added to the probe *R*-**1**, the fluorescence intensity at λ = 402 nm was substantially enhanced, and the fluorescence emission wavelength was red-shifted (from 389 nm to 402 nm). In the same case, when D-Lys was added, the fluorescence intensity at λ = 389 nm was reduced, the relative change was not significant (Figure 1b), and the value of the enantioselective fluorescence enhancement ratio was calculated as 31.27: [ef = |(*I_L_* − *I*_0_)/(*I_D_* − *I*_0_)|, where *I*_0_ refers to the fluorescence intensity of the fluorescent probe without the addition of the guest molecule, and *I_L_* and *I_D_* refer to the fluorescence intensity after the addition of the L-configuration and D-configuration guest molecules]. Therefore, it can be concluded that probe *R*-**1** showed excellent specific recognition of L-Lys (20 fold equivalent) in the chromatographic CH_3_OH system.

We further investigated the selective fluorescence recognition of D-*/*L-Lys by the chiral sensor *R*-**1** (2.0 × 10^−5^ mol/L chromatography CH_3_OH in mixed solution) for the enantiomeric isomers. Referring to Figure 2a,b, the fluorescence titration spectra of *R*-**1** with D-Lys and L-Lys verified that with the increase of L-Lys concentration (100-fold equivalent), the fluorescence intensity of probe *R*-**1** at λ = 411 nm was enhanced to 2.41 times of the initial intensity, and the emission wavelength was red-shifted from 389 nm to 411 nm; in the same case, with the addition of D-Lys (100-fold equivalent), the fluorescence intensity at λ = 389 nm showed a decreasing trend relative to the initial intensity, and the emission wavelength did not change. The analysis of the titration plot of D-*/*L-Lys shows that the fluorescence intensity of D-Lys shows a linear decrease, while the fluorescence intensity of L-Lys shows a linear increase, R = 0.9735(as shown in Figure 2c). In order to verify the stability of the complex between L-Lys and *R*-**1**, the saturation curve of L-Lys titration was performed, (Figure 2d) whose complex stability constant K_a_ = 6.47 × 10^6^ M was calculated by nonlinear fitting.

Therefore, it is further speculated that this is due to the unique spatial structure inside the *R*-**1** molecule synthesized based on the axial chirality of *R*-BINOL, which has different intensities of recognition for different configurations of lysine guest.

#### 3.2.2. Enantiomeric Excess’s Studies

We further investigated the composition of the lysine enantiomers. Probe *R*-**1** was made to mix with lysine at different ee [21,22] values, and fluorescence response measurements were performed. (Figure 3). It is seen that the fluorescence intensity gradually increases with increasing excess of L-Lys.

Referring to Figure 3, the fluorescence of *R*-**1** at 389 nm can show a linear increasing relationship with increasing enantiomeric excess (ee > 0), and the results of these curves can be obtained to identify the enantiomeric combination of amino acids. The trend of fluorescence intensity of probe *R*-**1** on lysine enantiomers confirms that probe *R*-**1** can be used to identify the enantiomeric combination of lysine. It is thus speculated that the molecular structure of the *R*-type compound *R*-**1** forms a space more suitable for L-Lys to approach the recognition site and form *R*-L complexes, thus achieving the selective recognition of lysine enantiomers.

#### 3.2.3. Study of Reaction Mechanism

In order to better investigate the recognition mechanism of lysine and fluorescent probe *R*-**1**, an ^1^H NMR spectroscopic titration test was performed by weighing 6.5 mg of *R*-**1** dissolved in DMSO and L-Lys dissolved in D_2_O (lysine is only soluble in water), adding 1.0 eq. of L-Lys first for spectroscopic scanning, and then 2.0 eq. of L-Lys for spectroscopic scanning (Figure 4). The analysis of the hydrogen spectrum spectra revealed that several sets of signal peaks of the probe *R*-**1** had different trends of chemical shift changes after the addition of L-Lys. Among them, since Ha is a reactive hydrogen atom in the hydroxyl group, it may combine with water after dropping into water, making its signal peak at 8.4 ppm disappear. Two Hb on the triazole group act as hydrogen bond donors and can form intermolecular hydrogen bonds with a pair of solitary pairs of electrons in the amino group at the L-Lys terminal [23], causing its chemical shift to move from 7.96 ppm to 7.77 ppm, a change of 0.19 ppm. Since the Hb participates in the formation of intermolecular hydrogen bonds. The chemical shift of Hd also changed from 2.82 ppm to 2.65 ppm, which changed by 0.17 ppm. The chemical shift of Hc was slightly shifted to the high field region, from 5.44 ppm to 5.36 ppm, a change of 0.08 ppm. The signal peak on the naphthalene ring did not change significantly, and the corresponding analysis can be concluded that the change mainly occurred in the triazole group.

According to the structural analysis diagram in Figure 4b, the strong electron-giving ability of the lone pair of electrons of the N atom in the amino group at the end of L-Lys enhances the electron-absorbing ability of Hb on the triazole group, which increases the electron transfer within the molecular structure and causes the intermolecular charge transfer effect (ICT) [24,25,26,27,28] to occur, resulting in the phenomenon of red-shift of emission wavelengths and fluorescence intensity enhancement. According to the titration diagram Figure 2b, the emission wavelength increased by 22 nm from 389 nm to 411 nm, while the fluorescence intensity enhanced by 2.41 times from 2548 to 6148. No significant wavelength and fluorescence intensity changes were found during the titration of D-Lys according to the titration diagram Figure 2a. The literature revealed that most of the fluorescence identifications that produced red-shift or blue-shift phenomena had an ICT mechanism generated, and the identification of lysine can be distinguished by this red-shift phenomenon and the corresponding fluorescence intensity change.

#### 3.2.4. Complexation Ratio Studies

As shown in Figure 5, to determine the complexation relationship between probe *R*-**1** and L-Lys, we performed a complexation ratio determination experiment of probe *R*-**1** and L-Lys. When performing the test, the total concentration of the mixed solution of probe *R*-**1** and L-Lys was kept at 2.0 × 10^−5^ M. The results showed that the molar fraction of [L-Lys]*/*([*R*-**1**] + [L-Lys]) reached the maximum value when the molar fraction of [L-Lys] was 0.5, indicating that probe *R*-**1** was bonded to L-Lys in the form of a 1 + 1 complex. Thus, it is known that the H atoms on both triazole groups of the *R*-**1** structure form an intermolecular hydrogen bonding force with the N atom at the L-Lys end group.

#### 3.2.5. Study of Specifically Reaction Lysine

Through experiments, such as nuclear magnetic titration and complexation titration, it was found that the identification group in the synthesized probe was the triazole group. Hb of C5 on the triazole group binds to the lone pair of electrons on N atoms in an amino group at the L-Lys end group position. Then, in terms of the structure of the eighteen pairs (thirty-six) of amino acids we identified, there are four amino acids with amino groups at the end base, namely asparagine, glutamine, arginine, and lysine. Their structures are:
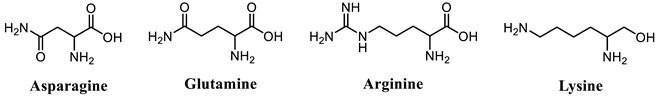


(1) Through the structural analysis of the four amino acids, although the end bases of Asp, Glu, and Arg also have amino groups, after the end base amino is connected to the carbonyl group and imine, the lone pair electron of the N atom in the amino group forms p-л conjugation with the carbonyl group and imine, which weakens the electron-giving ability of the lone pair electron. The structure of Lys is just an ordinary alkyl chain, and the ability to give electrons is relatively strong. (2) Through structural analysis, it can be seen that the structures of Asp, Glu, and Arg have greater steric hindrance than the structure of Lys, and Lys has only one alkyl chain and smaller steric hindrance. In summary, this probe only has fluorescence recognition for Lys.

#### 3.2.6. Recognition of Enantiomers *S*-**1** and *R*-**1**

In the same test environment, a series of tests were also carried out on the enantiomer *S*-**1** of *R*-**1**. According to Figure 6, both probes show the same fluorescence response with specific recognition of lysine (as shown in the Appendix A for specific test results).

#### 3.2.7. Fluorescence Titration of *R*-**1** and *S*-**1** by L-Lys

The results of titrating *R*-**1** and *S*-**1** with L-Lys are shown in Figure 7. When the titration is started with L-Lys, the fluorescence emission at λ = 389 nm is enhanced, and the emission wavelengths are accompanied by a significant redshift. However, the fluorescence enhancement rate of *R*-**1** is higher than that of *S*-**1**, indicating that its recognition is selectively differentiated. According to the linear Benesi–Hildebrand equation, the fluorescence [*I*_0_*/(I* − *I*_0_*)*] measured at 389 nm is a function of 1*/*[L-Lys], showing a good linear relationship (R > 0.9900). After the titration, the quantum yield of *R*-**1** and *S*-**1** increased from 4.40% to 5.39%, while the fluorescence lifetime of *R*-**1** and *S*-**1** increased from 2.02 ns to 3.20 ns.

#### 3.2.8. Fluorescence Titration of *R*-**1** and *S*-**1** by D-Lys

The results of titrating *R*-**1** and *S*-**1** with D-Lys are shown in Figure 8. When the titration is started with D-Lys, the fluorescence emission at λ = 389 nm does not change significantly, and the fluorescence intensity decreases slightly. Similarly, the fluorescence intensity of *R*-**1** varies more than *S*-**1**, indicating that its recognition can be selectively distinguished. According to the linear Benesi–Hildebrand equation, the fluorescence [*I*_0_*/(I* − *I*_0_*)*] measured at 389 nm is a function of 1*/*[D-Lys], showing a good linear relationship (R > 0.9900). After the titration, the quantum yield of *R*-**1** and *S*-**1** decreased from 4.40 % to 4.30 %, while the fluorescence lifetime of *R*-**1** and *S*-**1** decreased from 2.02 ns to 1.20 ns.

## 4. Conclusions

In summary, a novel chiral BINOL fluorescent probe linked by a triazole group was synthesized by click reaction and nucleophilic substitution reaction when 36 amino acids (18 amino acids of different configurations) were added to the probe, which can specifically identify and distinguish different configurations of lysine. Through the study of different configurations of lysine, it was found that L-Lys notable enhanced the fluorescence intensity and red-shifted the emission wavelength, and this phenomenon was attributed to the stronger electron-giving ability of the lone pair of electrons of the N atom in the amino group at the L-Lys end position, which enhanced the electron-absorbing ability of H on the triazole group and improved the electron transfer ability within the molecular structure, causing the intramolecular charge transfer effect (ICT). Among the 36 amino acids added, there are 4 amino acids with amino groups at the end base, namely Asp, Glu, Arg, and Lys. Due to lysine having a simple alkyl chain, it has a stronger ability to give electrons and less steric hindrance, so the probe has strong fluorescence recognition with lysine. We also explored the enantiomeric *S*-**1** (as shown in Appendix A) and found that the distinguished recognition of chiral amino acids has the same recognition preference. As amino acids are essential to the human body, different activities exist for different configurations of amino acids, so chiral fluorescent sensors are used to recognize different configurations of amino acids as one of the necessary methods.

## Data Availability

Not applicable.

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
