# Peer review of "Enantioselective Recognition of L-Lysine by ICT Effect with a Novel Binaphthyl-Based Complex"

_micromachines, 2023, doi:10.3390/mi14030500_

Round 1
Reviewer 1 Report (Previous Reviewer 2)
The author has synthesized fluorescent sensors with different stereochemistry, which respond selectively only to L-lysine.
Why the difference in fluorescence intensity against D and L?
Why does the stereochemistry of the sensor not affect the D or L of amino acids?
Also, can these phenomena be called enantioselective recognition?
Author Response
Please see the attachment.

Reviewer 2 Report (Previous Reviewer 3)
The manuscript represents the synthesis of two new enantiomeric BINOL-based sensors R-1 and S-1, and their enantiomeric recognition studies toward 18 pairs of amino acid enantiomers. The manuscript has improved, and the authors performed repeating experiments to confirm the behavior of the sensors. However, the manuscript contains further mistakes, which should be addressed.
Important references (listed below), which were involved in the original manuscript, but were deleted, should be cited in the appropriate points of the Introduction (in the 1st and 3rd paragraphs).
Pu L. Enantioselective fluorescent sensors: a tale of BINOL. Accounts of Chemical Research, 2012, 45 (2): 150-163.
Zhang X, Yin J, Yoon J. Recent advances in development of chiral fluorescent and colorimetric sensors. Chemical Reviews, 2014, 114 (9): 4918-4959.
Pu L. Simultaneous determination of concentration and enantiomeric composition in fluorescent sensing. Accounts of Chemical Research, 2017, 50 (4): 1032-1040.
R-1: "4.96 (d, J = 12.4 Hz, 1H), 4.89 (d, J = 12.5 Hz, 1H)" in line 101 and supplementary sections 1.4, 2.2: should be given as an AB quartet [e.g.: "4.93 (ABq, ΔδAB = 0.06, J = 12.5 Hz, 2H)"]
S-1: "4.93 (q, J = 12.5 Hz, 2H)" in line 109 and "4.94 (d, J = 12.4 Hz, 1H), 4.88 (d, J = 12.5 Hz, 1H)" in supplementary sections 1.5, 2.3: should be given in the same way, as an AB quartet (similarly to R-1). The other 1H NMR signals of S-1 in manuscript 2.1.3. should be the same as in supplementary sections 1.5 and 2.3.
The authors should correct (duplicate) the listed proton numbers in the cases of R-1, S-1, R-b and R-c, because it seems that only the half parts of the molecules were considered. Please, check these in the manuscript and the supplementary material (characterizations and 1H NMR spectra). (Of course, because of the duplication, the above mentioned two ABq-s for R-1 and S-1 will be 4H instead of 2H.)
Line 37, "basic": should not be underlined
Line 63, "drugs": no drugs were mentioned in this work, did the authors mean "chemicals" instead of it?
The systematic names of R-c (in lines 72, 90 and supplementary sections 1.3, 1.4) and S-c (supplementary section 1.5) are not correct, they should be checked with ChemDraw, and correct accordingly.
Line 84: solvent mixture used for the column and the type of column (silica gel?) are missing
Line 90: "(R-c)" should be after the systematic name, before the amounts
Lines 90-91: there is a mistyping in "2-azidoacetate"
Lines 95 and 97: solvents should be given with names not with abbreviations
Line 97: the type of column (silica gel?) is missing
Line 172: according to Figure 2b, the fluorescence enhancement is larger than 1.50 times
Row 181: "was perfomed" → "was calculated"; the method should be given after "was calculated" (by the Benesi-Hildebrand method or by nonlinear fitting?)
Caption of Figure 2: "(e)" →"(d)"
Lines 218 and 250: "intramolecular" → "intermolecular"
Line 220: "Since Hb forms The chemical shift of Hd was …": the authors should correct the sentence.
Line 229: "of chemical shift red shift": it is not correct, the authors should use e.g., "of red shift of emission wavelengths" instead of it
Lines 231-232: data ("15 nm from 390 nm to 405 nm", "1.7 times from 1958 to 3243") are given according to the old Figure 2b. The data should be updated to the present (new) Figure 2b, and should be in accordance with the data in lines 172-174.
Line 234: "Figure 2(c)" → "Figure 2(a)"
Figure 5, x axis label: [I0] should change to [R-1]
Line 260: "Glutarnine" → "Glutamine"
Caption of Figure 7: the added equivalents should be listed for (a) and (b)
Caption of Figure 8: the added equivalents should be listed for (a) and (b)
Lines 287 and 305: "absorbance" → "fluorescence"
Lines 288 and 306, Figure 7c, Figure 8c: "1/[M]" → "1/[Lys]" (because M usually used for metal ions)
Lines 289, 290, 307 and 308: "1" should be given as "R-1" or "S-1" (which was measured)
The UV-Vis titration measurements of R-1 and S-1 with L-Lys and D-Lys should be put to the end of the Supplementary material.
Round 2
Reviewer 1 Report (Previous Reviewer 2)
I did not still completely understand, but the author responded my comments based on the results.
Author Response
Please see the attachment.

Reviewer 2 Report (Previous Reviewer 3)
R-1: "ABd" → "ABq" in line 105 and supplementary sections 1.4, 2.2.; S-1: "ABd" → "ABq" in line 122 and supplementary sections 1.5, 2.3. (Because the signals are AB quartets due to the diastereotopic nature of –CH2– protons.)
R-1, S-1, R-b and R-c: all of the listed 1H NMR signals are correct. I only meant that hydrogen numbers (1H, 2H, etc.) are the half of neccessary, because the molecules are symmetrical, and only hydrogens in the half part were counted. That is why I suggested to duplicate the hydrogen numbers for each signals, for example (line 103): "7.88 (s, 1H), 7.83 (d, J = 7.9 Hz, 1H)", etc. → "7.88 (s, 2H), 7.83 (d, J = 7.9 Hz, 2H)", etc. The authors should check and correct these in the manuscript and the supplementary material (characterizations and 1H NMR spectra). (R-a is correct.) (Similar symmetrical BINOL derivatives can be seen in J. Mol. Struct., 2023, 1276, 134793.)
Line 85 and supplementary section 1.3: solvent mixture used for the column is missing
Lines 194-196, there is a mistake in the correction, the authors should make this change: "L-Lys titration was calculated by nonlinear fitting, and (Figure 2d) whose complex stability constant Ka = 6.47×106 M was performed." → "L-Lys titration was performed, and (Figure 2d) whose complex stability constant Ka = 6.47×106 M was calculated by nonlinear fitting."
The software or a reference for the equation, which was used for the nonlinear fitting should be given in "Materials and methods".
Line 236: "within Hb molecules"? The meaning of this is not clear.
Line 237: "changed to 0.17 ppm" → "changed by 0.17 ppm"
Figures 7c and 8c, x axis label: "1/[M]" → "1/[Lys]"
Lines 327 and 328: "increased" → "decreased"
The UV-Vis titration measurements in the Supplementary material should be Figure 3 instead of Figure 1.
Figure 2(d) and Suppl. Figure 1(e): the red lines are not fitted curves, so "nonlinear fitting" in the figure captions should not be written
It is disturbing that there is no explanation for the unusual behavior of the sensors (both R-1 and S-1 recognize L-Lys), however, the authors performed repeating / additional experiments to exclude the interchange of the titrant amino acid enantiomers and the presence of contaminants.
Round 3
Reviewer 2 Report (Previous Reviewer 3)
After the following small corrections, the manuscript (and suppl. material) can be accepted for publication.
Line 70: "Nonlinear" → "nonlinear"; "by" → "was"
Lines 96 and 115: "Sodium" → "sodium"
Line 238: "The Hb participates" → "Since the Hb participates"
Line 310: "[Lys]" → "[L-Lys]"
Line 330: "[Lys]" → "[D-Lys]"
Line 354: "mechanism" → "preference"
Suppl. material section 2.3: "ABd" → "ABq"
This manuscript is a resubmission of an earlier submission. The following is a list of the peer review reports and author responses from that submission.
Round 1
Reviewer 1 Report
The authors here constructed a novel BINOL derivative as a fluorescent probe for enantioselective recognition of L-Lys and D-Lys. The enantioselective study with different instrumentation remains a hot topic in areas of imaging and organic synthesis. Overall, the manuscript was well written and authors were about to tackle both chemoselectivity and enantioselectivity of the amino acid, which still remained little explored. I would like to recommend it for publication if the authors can address the following concerns and issues.
1. The synthesis and characterization of the S enantiomers are also required. If they were characterized in the supporting material that I didn't received so far.
2. In total there're often 18-19 amino acids, but it seems the authors only tested 16 of them including Lys. Could you please justify the missed ones (Trp, Leu, Ile)? Otherwise it's hard to claim the chemoselectivity of the probe for Lys only.
3. Sometimes due to the vendor's issue or long time storage, Lys was not pure enough and caused false positive selectivity thought the chance was very very low. The purity check of the L-Lys and D-Lys is also recommended to confirm the truth of the selectivity, which is not required to report but suggested.
4. Detailed fluorescence study procedure (incubation volume, incubation time, and stability of the signal) is needed for the report.
5. In Figure 4, I anticipate that different NMR solvents were used in the four NMR spectra, please specify them.
6. Did the authors ever screened other solvents for the fluorescence responsive study of the probe R-1 and S-1?
7. Line 105, D/L-cytosine probably should be D/L-cysteine?
8. Multiple synthesis steps are needed from the BINOL starting material, it's not fair to state as "A novel triazole fluorescent sensor was efficiently synthesized by click reactions using binaphthyl phenol as a substrate with a yield of 85%" in the Abstract, please rephrase it either from the alkyne or report the total synthesis yield from the binaphthyl phenol as the starting material.
9. The intention or the starting point of the research was not well elaborated. Why did the author designed the structure? or Just randomly screened a series of BINOL derivatives?
I guess some of the data that were needed had been included in the supporting material as the manuscript described. Hopefully, next time the supporting material will be provided when it is resubmitted.
Author Response
Dear reviewer,
Thank you very much. I am very grateful to your comments for the manuscript. According with your advice, we amended the relevant part in manuscript. Some of your questions were answered below. In order to make it easier for you to view the revisions, the specific number of pages and lines of the revised parts of the manuscript is marked in red in the attachment.
Xiaoxia

Reviewer 2 Report
This paper reported the synthesis a novel Binaphthyl-based complex which exhibited the enantioselective recognition of L-Lysine by ICT mechanism. The method is simple, and the research is relatively systematic. However, I do not see any description or discussion of the essential question of why it recognizes only lysine specifically. However, there is no description or discussion of the essential question of why compound 1 recognizes only Lys specifically. I would like to make a decision after this part is described.
Author Response

(The authors gave the same response as above.)

Reviewer 3 Report
This manuscript represents the synthesis and enantiomeric recognition studies of two new enantiomeric BINOL-based sensors R-1 and S-1 toward 15 pairs of amino acid enantiomers.
Several reported BINOL and other types of sensors, of which both enantiomers were studied, showed practically the same fluorescence changes for (R)-host–(S)-guest and (S)-host–(R)-guest complexation as an evidence for chiral recognition (e.g. in Ref. 4). This is because (R)-host–(S)-guest and (S)-host–(R)-guest complexes are in enantiomeric relationship with the same fluorescence quantum yield.
The topic presented here is valuable, but the basic problem is that the results do not show the above pattern, because R-1 recognizes L-Lys (Figure 1b), and S-1 also recognizes L-Lys instead of D-Lys (Figure 6). What is the reason for this? The authors should repeat the experiments carefully to figure out what could be the problem.
The authors identified the interaction of R-1 and L-Lys as a 1:1 complexation, but the complex stability constant is missing. The authors should determine it. It is important to note that a 1:1 complexation gives a saturation curve during the titration up to the end-point, but the titration curve for L-Lys in Figure 2d is linear. Maybe, this is because the complexation is weak, and the addition of a larger amount (up to 400-500 equiv.) of L-Lys is nedded to get the saturation curve, which could be used for the determination of the stability constant by the Benesi-Hildebrand method. However, if the authors do not get this curve (which is characteristic for complexation), they should explore the reason for it and the type of interaction.
The ef value of 2.91 (row 126 and many times mentioned) can not be correct, because (IL-I0) is a large positive value, and (ID-I0) is a negative value with a very small absolute value. The authors should check this also. (In the abstract, the parentheses are also missing from the equation.)
Row 123: the authors should check the 397 nm wavelength value, because it is not consistent with Figure 1b, it seems to be 405 nm (which is mentioned later also).
Row 142: the authors meant IL/ID instead of ID/IL? They should check this.
Figure 3: the experiments should be repeated more precisely, because the starting points (at ee = 0%) are the same.
In section 3, the synthesis part should be discussed in one or two more sentences as usual, which also contain the citation of the last reported compound in the reaction scheme. Scheme 1 should be put here, not the end. In Scheme 1 R-a, R-b, R-c should be used instead of a, b, c. These should be used in parentheses after the compund names in section 2.1. Compund names should be checked (e.g. with ChemDraw). The characterization of S-1 is missing in section 2.1, its specific rotation have to be given, and other spectroscopic data should be compared with those of R-1, and if the same, its enough to write that.
The details of the fluorescence measurements (preparation of solutions) should be listed in section 2 instead of other parts (the grade of the solvent should be defined here also, but later in the text it is enough to mention methanol or CH3OH).
The authors should check the whole manuscript carefully regarding:
- English grammar
- the terms, e.g. "alkaline" à "basic" (row 37); "binaphthyl phenols" à "binaphthols" (rows 50, 225); "a novel chiral fluorescent probes for binaphthol" à "novel chiral fluorescent probes based on binaphthol" (row 53); "different conformations of" lysine/amino acids à "different configurations of" lysine/amino acids (rows 18, 148, 227, 228, 236, 237); etc.
- unclear parts, e.g. "Suppose the body lacks or reduces one of them"; etc.
- in "λex", the "ex" should be in subscript in all cases
- row 121 (in section 3.1.1.): "ex" should be deleted from "λex="
- some of the references are redundant: e.g., 8 and 10; 1 and 14; 12 and 24; 2 and 27
Author Response
Dear reviewer,
Thank you very much. I am very grateful to your comments for the manuscript. According with your advice, we amended the relevant part in manuscript. Some of your questions were answered below. In order to make it easier for you to view the revisions, the specific number of pages and lines of the revised parts of the manuscript is marked in red in the attachment.

Round 2
Reviewer 2 Report
The author responds in the response sheet, but I do not see any additions to the manuscript. Addition, the structure of 3-bromopropyne is incorrect yet.
Reviewer 3 Report
The authors suggest in their reply that "(2) If chiral C is closer to the recognition site, then the probe will recognize amino acids of opposite configuration. In the manuscript, the recognition site of our synthesized compound is slightly farther away from the chiral center, so the recognition situation may vary."
If the recognition site is far away from the chiral center in S-1 and R-1, what would be the effect which provides the differentiation between L-Lys and D-Lys? This is a contradiction. If the differentiation between L-Lys and D-Lys is based on complexation as determined by the authors, the two enantiomeric complexes R-1–L-Lys and S-1–D-Lys should give the same fluorescence quantum yield, and the same fluorescence changes, because they are enantiomers with the same physical properties (such as R-1 and S-1). So, it should be an anomaly that apparently both R-1 and S-1 recognize L-Lys.
However, it can be possible that the fluorescent changes are originated from a contaminant present in the L-Lys material, but not from L-Lys; therefore, the authors should check the optical (specific) rotations of L-Lys and D-Lys, compare with the literature values, record their NMR and UV–Vis spectra to see their purity. It is very important to explore what causes the anomaly (meaning that both R-1 and S-1 recognize L-Lys).
Characterization of S-1 (page 3): its specific rotation is given as -75 (instead of around -150), and R-1 has the value of +150. It is a very big difference (-75 vs. -150), meaning that S-1 contains a large amount of contaminants. The 13C NMR spectrum of S-1 also differs from that of R-1: 62.36 (in S-1) vs. corresponding signal 52.95 (in R-1), and S-1 contains plus two signals 29.67 and 13.99, which are not present in R-1. The authors should purify S-1, then redone the fluorescence experiments.
Row 17: there are missing parentheses in the equation. In row 162, the parentheses are correct in the equation. In row 17, the equation should be corrected according to row 162.
Another problem with "ef" that according to the equation in row 162, the "ef" is a negative value, because (ID-I0) is negative. So the authors should consider to write the negative value –76.57 or use the absolute value sign in the equation such as "ef = Ç€(IL-I0) / (ID-I0)Ç€". (The correction which is chosen by the authors should be applied where the equation or "ef" is present in the manuscript.)
In Scheme 1, R-a, R-b, R-c should be used instead of a, b, c.
